# Genomic and phenotypic stability of fusion-driven pediatric sarcoma cell lines

Merve Kasan[1,2,3,4], Florian H. Geyer[1,2,3], Jana Siebenlist[1,2,3], Martin Sill[1,5], Rupert Öllinger [6], Tobias Faehling [1,2,3], Enrique de Álava [7,8], Didier Surdez[9,19], Uta Dirksen [10], Ina Oehme [1,3,11], Katia Scotlandi [12], Olivier Delattre [9], Martina Müller-Nurasyid[13,14,15], Roland Rad[6,16,17], Konstantin Strauch[13,14,15], Thomas G. P. Grünewald[1,2,3,4,18] & Florencia Cidre-Aranaz [1,2,3,4] ✉

Human cancer cell lines are the mainstay of cancer research. Recent reports showed that highly mutated adult carcinoma cell lines (mainly HeLa and MCF-7) present striking diversity across laboratories and that long-term continuous culturing results in genomic/transcriptomic heterogeneity with strong phenotypical implications. Here, we hypothesize that oligomutated pediatric sarcoma cell lines mainly driven by a fusion transcription factor, such as Ewing sarcoma (EwS), are genetically and phenotypically more stable than the previously investigated adult carcinoma cell lines. A comprehensive molecular and phenotypic characterization of multiple EwS cell line strains, together with a simultaneous analysis during 12 months of continuous cell culture show that fusion-driven pediatric sarcoma cell line strains are genomically more stable than adult carcinoma strains, display remarkably stable and homogenous transcriptomes, and exhibit uniform and stable drug response. Additionally, the analysis of multiple EwS cell lines subjected to long-term continuous culture reveals that variable degrees of genomic/transcriptomic/phenotypic changes among fusion-driven cell lines, further exemplifying that the potential for reproducibility of in vitro scientific results may be rather understood as a spectrum, even within the same tumor entity.

Cancer cell lines have been instrumental in biomedical progress for many decades[1–3]. In 2018 and 2019, respectively, Ben-David et al. and Liu et al. showed that highly mutated adult carcinoma cell lines present striking diversity across laboratories and that long-term continuous culturing results in genomic/transcriptomic heterogeneity with phenotypical implications, including changes in drug sensitivity[4], doubling time, and response to a specific perturbation[5], which challenged the general reproducibility of scientific results based on human cancer cell lines. However, to which extent these observations can be generalized to every cancer cell line remains to be explored.

The multi-omics study by Liu et al. showed a substantial heterogeneity between different variants of the first human-derived cancer cell line, HeLa (cervix carcinoma)[6], mainly between the most commonly used variants HeLa-CCL2 and HeLa-Kyoto. Interestingly, Ben-David et al. reanalyzed the genomic data (whole exome sequencing) of 106 cancer cell lines provided by the Broad and the Sanger Institutes and showed a significant diversity in allelic fraction for somatic variants in this panel of cell lines. Notably, this panel mainly consisted of hematopoietic/lymphoid and adult carcinoma cell lines and only included a single EwS cell line (CADO-ES1), which was not further investigated. Among those adult carcinoma cell lines, the authors specifically focused on the estrogen receptor-positive adult breast carcinoma cell line MCF-7 for a cross-laboratory analysis and demonstrated crucial genomic, transcriptomic, and phenotypical diversity.

They additionally verified their findings in a panel of adult carcinoma cell lines including (except for a single pediatric hepatoblastoma cell line HepG2-A) mostly adult-type carcinoma cell lines, all of which are not driven by a single mutation, such as the chimeric oncogenic transcription factor (COTF) found in EwS.

In this study, we hypothesize that oligo-mutated pediatric sarcoma cell lines driven by a COTF, such as Ewing sarcoma (EwS)[7] are genetically and phenotypically more stable than the previously investigated adult carcinoma cell lines[4,5]. By performing extensive genomic, epigenomic, transcriptomic, and phenotypic analyses on multiple oligo-mutated pediatric sarcoma cell line strains in strict comparison with the two carcinoma-derived cell lines, we observe that EwS cell lines are genetically and phenotypically more stable than the previously investigated adult carcinoma cell lines. In addition, our results highlight that when subjected to long-term culture conditions, individual cell lines from the same cancer entity may display a variable degree of evolution, further indicating that the reproducibility of cell line-based scientific results strongly depends on the given cancer cell line.

## Results

To first test whether fusion-driven sarcoma cell lines are clonal or genetically unstable, we selected human A-673, one of the most widely used EwS cell lines, and compared 11 A-673 strains with five strains of human HeLa cervix cancer and five strains of human MCF-7 breast cancer collected from seven, three, and two different laboratories, respectively (Fig. 1a). Despite some of these strains had an undefined number of passages, they were considered serviceable for cell biology research. In this comparison, we included a newly purchased strain for each cell line, which was continuously cultured for 12 months, and examined at three different time points (corresponding to months 0, 6, and 12, hereafter referred to as m0, m6, m12) (Fig. 1a). To reduce empirical bias prior to our (epi)genomic, transcriptomic, and phenotypical analyses, we cultured all strains in the same cell culture conditions (see Materials and Methods section).

In the first step, we performed a cross-strain analysis of A-673, MCF-7, and HeLa cell lines and subjected each newly purchased cell line (m0) and its respective m6-cultured version to whole genome sequencing (WGS), which enabled us to monitor genetic evolution over time. Analysis of relative in-exon SNVs counts in cancer genes for A-673, HeLa, and MCF-7 after six months of continuous culture revealed general stability of A-673 strains as compared to HeLa and MCF-7 cells (Fig. 1b). To explore the differences in genome stability comparing COTF-driven strains to carcinoma strains in more detail, we employed WGS data and compared copy number alternations (CNAs) of A-673 and MCF-7 strains from different laboratories including our A-673_m0 and MCF-7_m0, those of the Cancer Cell Line Encyclopedia (CCLE), and an A-673 strain from the Ewing Sarcoma Cell Line Atlas (ESCLA). As displayed in Supplementary Fig. 1a, b, the A-673 strains generally presented a more stable genome compared to MCF-7 strains, as quantified by relative changes in copy numbers. An expansion of these analyses by exploring non-synonymous SNPs that affected the coding sequence and splicing regions for the 11 different A-673 strains (including two A-673 with genetic modifications) using Illumina Global Screening Arrays (GSA) revealed that 98.9% were shared by all strains (Fig. 1c), which drastically diverged from the only 35% of SNPs shared by all strains in MCF-7[4].

To investigate this discrepancy between adult carcinoma cell lines and oligo-mutated pediatric sarcomas at the transcriptional level, we compared the transcriptomic variation of these previously studied adult carcinoma cell lines with fusion-driven EwS cell lines. Specifically, we performed RNA sequencing (RNASeq) using NextSeq 500 (Illumina) on 11 A-673, five HeLa, and five MCF-7 strains. Principal component analysis (PCA) performed on the transcriptomic data from three biological replicates per cell line revealed that similar to the observations

made by Ben-David et al. and Liu et al.[4,5]., the strains of both carcinoma cell lines showed widespread transcriptomic diversity. However, our fusion-driven A-673 EwS strains clustered tighter than HeLa and MCF-7 carcinoma strains, even though the A-673 cluster contained two strains with genetic modifications, and both carcinoma cell lines had relatively smaller sample sizes (Fig. 1d). To analyze this variability specifically within each lineage, we conducted independent DGEA and PCA on the 11 A-673, five HeLa, and five MCF-7 strains and computed the variance percentages across each cell line's strains. We thus observed that the A-673 strains demonstrated a 2- to 3-fold smaller variance compared to the carcinoma cell lines, again highlighting the higher stability of the A-673 strains, even considering the larger sample size in the A-673 collection (Supplementary Fig. 1c). These observations were additionally confirmed by analyzing the coefficient of variation (CV) of gene expression for each cell line (Fig. 1e).

To study this phenomenon in more detail, we compared each cancer entity with the two strains with the highest variance (A-673_7 and A-673_3 vs. HeLa_5 and HeLa_3 vs MCF-7_5 and MCF-7_3). Strikingly, we observed over 60 times more differentially expressed genes (DEG) defined as |fold change (FC)| > 1, Benjamini-Hochberg (BH) adjusted $P < 0.01$ (380 transcripts; 39 up-, 341 down-regulated) in the HeLa strains and 20 times more DEG (108 transcripts; 57 up-, 51 down-regulated) in the MCF-7 strains as compared to the A-673 EwS strains (5 transcripts; all up-regulated) (Fig. 1f and Supplementary Fig. 1d). We additionally combined our transcriptomic data with that of Liu et al. and observed a specific clustering of our HeLa strains with their HeLa-CCL2 strains, indicating a likely common origin (Fig. 1g). Moreover, we observed a remarkably higher degree of heterogeneity among HeLa strains than among A-673 strains (Fig. 1g and Supplementary Fig. 1e). Of note, considering this heterogeneity between HeLa-CCL2 and Kyoto strains described by Liu et al. and here, it is conceivable that the inclusion of HeLa-Kyoto in our panel of cells (Fig. 1a) would have resulted in an even more substantial difference when compared to fusion-driven A-673.

Next, we compared the expression profiles of the newly purchased cell lines (m0) for each cancer entity with their m12 derivates. Consistent with the results observed in the cross-laboratory comparison, we observed a significantly greater variation in global gene expression ($P < 0.0001$, two-sided Wilcoxon signed-rank test) in HeLa and MCF-7 cells compared to A-673 (median $\log_2 FC_{A-673} = 0$, $-4.25 < \tilde{X} < 4.22$; median $\log_2 FC_{HeLa} = 0.47$, $-3.39 < \tilde{X} < 16.89$; median $\log_2 FC_{MCF-7} = 0.47$, $-3.46 < \tilde{X} < 15.81$) (Fig. 1h and Supplementary Fig. 1f).

To evaluate the potential phenotypical impact of these genomic and transcriptomic changes, we compared the drug responses of fusion-driven EwS cells with those from highly mutated adult carcinomas. Thus, we subjected 11 A-673 EwS strains (including two A-673 with genetic modifications, Fig. 1a), five HeLa cervical cancer strains, and five MCF-7 breast cancer strains to a drug screening consisting of a selection of 10 active compounds addressing non-redundant functional pathways, which encompassed the same drugs used in Ben-David et al. [4]. The obtained dose-response curves were used to compute the area under the curve (AUC) for each compound and to determine the respective Euclidean distances (ED) between sensitivity profiles of a given cell line to the global AUC-mean across cell lines. In agreement with our previous findings, the strains of both adult carcinomas exhibited a significantly higher degree of drug response variability than those of EwS for all screened compounds (Fig. 1i, $P < 0.005$, one-sided Wilcoxon signed-rank test). To confirm the extensive homogeneity in drug response of the fusion-driven EwS cells as compared to carcinoma cell lines, we additionally performed a Spearman's correlation test among each cell line's strains and once again observed that EwS strains showed a higher similarity than carcinoma cell lines ($\tilde{X}_{\text{Spearman's } \rho \text{ A-673}} = 0.94$, $0.95 < \tilde{X} < 0.93$; $\tilde{X}_{\text{Spearman's } \rho \text{ HeLa}} = 0.87$, $0.91 < \tilde{X} < 0.83$; $\tilde{X}_{\text{Spearman's } \rho \text{ MCF-7}} = 0.88$, $0.92 < \tilde{X} < 0.85$) (Fig. 1j).

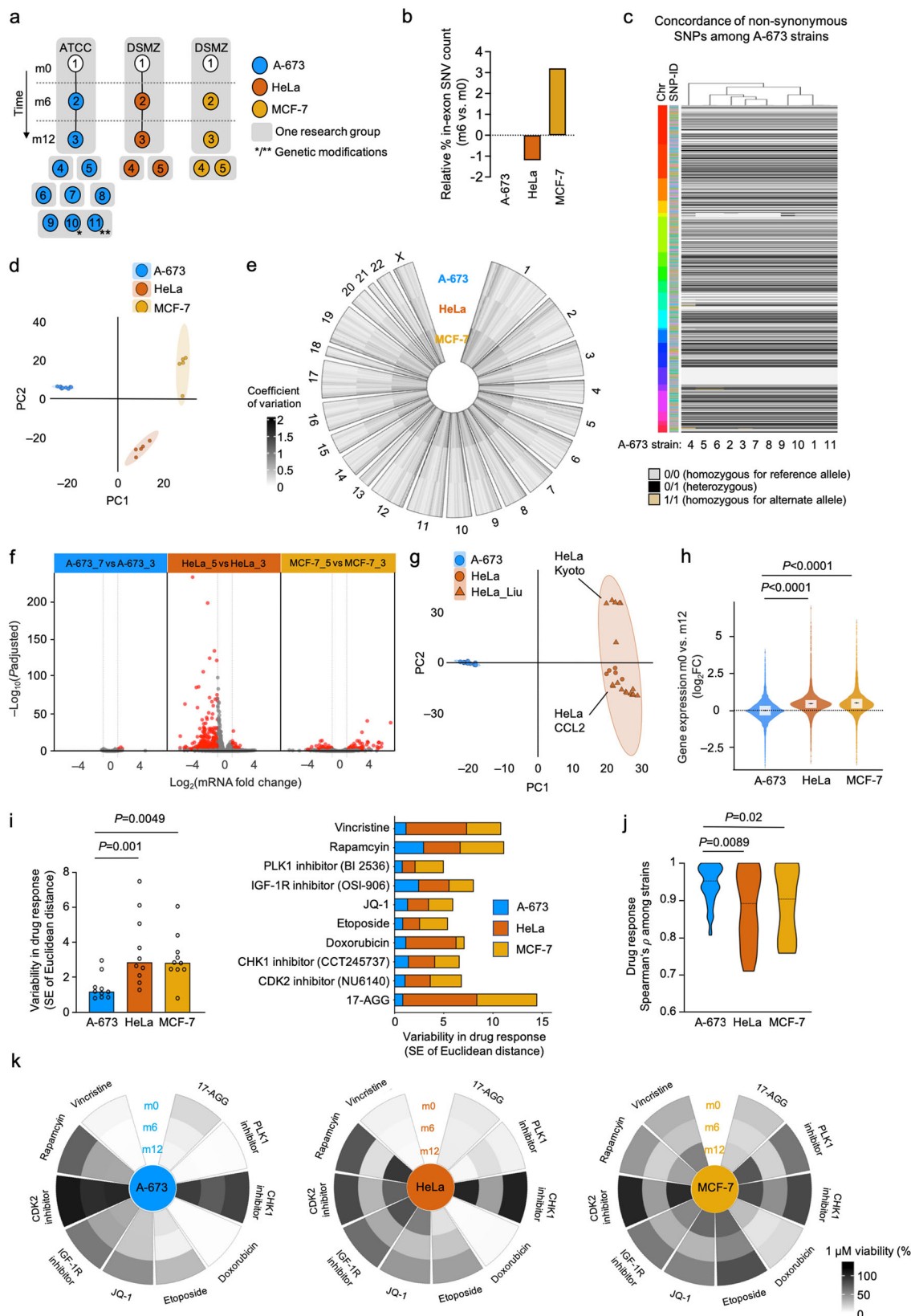

Further, we studied the effect of continuous culture over 12 months on the potential evolution in drug sensitivity. Therefore, we exposed newly purchased A-673, MCF-7, and HeLa cells (m0) to the drug library, and then again at two additionally predefined time points after continuous culture (m6 and m12). In agreement with previous findings, we detected a remarkably stable phenotype of A-673 after 6 and 12 months in comparison with HeLa and MCF-7 cell lines, measured as raw viability at a single concentration of each compound (1 μM) (Fig. 1k).

Finally, to expand our understanding of the scarcity of genomic and phenotypic cell line evolution in the context of EwS, we sought to analyze how our findings in A-673 cells (as one of the most widely used cell lines in EwS research) would compare to other EwS cell lines. Thus, we

**Fig. 1 | Fusion-driven pediatric sarcoma cell line strains exhibit exceptional genomic, transcriptomic, and phenotypic stability compared to adult carcinoma strains. a** Newly purchased A-673 EwS, HeLa cervix carcinoma and MCF-7 breast carcinoma wild type cell lines (A-673_1, HeLa_1 and MCF-7_1) were kept in culture for six months (m6; A-673_2, HeLa_2 and MCF-7_2) and twelve months (m12; A-673_3, HeLa_3 and MCF-7_3). In addition, multiple strains for each cell line were collected from seven, three, and two laboratories, respectively, and arbitrarily numbered as A-673_4 to A-673_9, HeLa_4 to HeLa_5, and MCF-7_4 to MCF-7_5. Single-cell clones derived from A-673 cell lines with a neutral manipulation (*, A-673_10) or an inducible shRNA construct targeting its EWSR1::FLI1 translocation (**, A-673_11) were included. ATCC, American Type Culture Collection, DSMZ (German Collection of Microorganism and Cell Cultures). **b** Relative in-exon SNVs counts in cancer genes for A-673, HeLa, and MCF-7 after six months of continuous culture, using the respective initial time point (m0) values as reference (m6 vs. m0). **c** Heatmap comparing the status (homozygous for reference allele, alternate allele, or heterozygous) of non-synonymous SNPs in 11 A-673 strains. The left color bar depicts chromosomes; the right color bar depicts different SNP-IDs ($N = 1,599$).
**d** Transcriptomic PCA of 11 A-673, five HeLa, and five MCF-7 cell line strains

($N = 10,256$ transcripts). **e** Circle plot depicting coefficient of variation (CV) of expressed genes per chromosome (top 60% quantile) for all A-673, HeLa, and MCF-7 cell line strains. **f** Volcano plot of DEG resulting from comparing the two A-673, HeLa, and MCF-7 strains with the highest variance (A-673_7 vs A-673_3, HeLa_5 vs HeLa_3 and MCF-7_5 vs MCF-7_3). The red dots denote significantly differentially expressed genes (BH adjusted $P < 0.01$; $|FC| > 1$). **g** Combined transcriptomic PCA of our A-673 and HeLa datasets and that of Liu et al. (HeLa_Liu) ($N = 13,569$ transcripts). **h** Relative gene expression of A-673, HeLa, and MCF-7 cell lines after long-term culture for twelve months (m12 vs m0). Outer violin curves denote the kernel density. Boxplots display the interquartile range and the mean, two-sided Wilcoxon signed-rank test. **i** Left, collective variability in drug response of all A-673, HeLa, and MCF-7 cell line strains depicted as the standard error of ED, each compound is shown as a black circle, one-sided Wilcoxon signed-rank test. Right, standard error of ED for each specific screened compound. **j** Spearman's $\rho$ of drug response across 11 A-673, five HeLa, and five MCF-7 cell line strains. The dotted black line shows the median (one-sided Wilcoxon rank-sum test). **k** Raw viability of A-673, HeLa, and MCF-7 cell lines subjected to each compound (1 μM) included in the drug screening after 0, 6, and 12 months of continuous long-term culture (m0, m6, and m12).

newly purchased four additional EwS cell lines (MHH-ES1_m0, SK-ES-1_m0, SK-N-MC_m0, and TC-71_m0) and propagated them for 12 months (Fig. 2a). We first performed genomic and epigenomic analyses and subjected our samples to Illumina GSA and MethylationEPIC BeadChip arrays, respectively. Interestingly, while all EwS cell lines remained relatively stable, we observed that when compared to the prototypical A-673 cell line, the remaining EwS cell lines presented a gradient of variability when analyzing both their non-synonymous SNP alterations and their differentially methylated CpG sites over time (Fig. 2b, c). For instance, while a median of 99.6% (range 99.3%–99.8%) of the in-exon SNPs were shared after 12 months of continuous culture by each cell line, we could observe relatively less stable cell lines such as A-673, and MHH-ES-1, and remarkably stable cell lines such as TC-71 (Fig. 2b), whereas SK-ES-1 and A-673 displayed less stability at epigenetic level (Fig. 2c). Of note, the relatively low number of variable ns-SNPs found in A-673 strains appeared to affect random genes and to be not enriched in specific pathways or biological processes (Supplementary Data 2). Only one known EWSR1::FLI1-signature gene (UNC5 family of netrin receptors, *UNC5B*)[8] was affected. In addition, when we tested the consistency of differentially methylated CpG sites across all EwS cell lines particularly located at promoter regions, we found only 1% overlap (corresponding to 51 promoter regions) (Supplementary Fig. 2a and Supplementary Data 3). This observed genomic and epigenomic variability in the degree of evolution over time was further detected at the transcriptional level, as shown by the proportion of significant DEG of each EwS cell line when compared to their m12 derivate (Fig. 2d). Indeed, TC-71 showed the least transcriptional changes over time, while SK-ES-1 exhibited the highest number of DEG after 12 months of continuous culture (219 transcripts; 99 up- and 120 down-regulated, which represented a 50% increment relative to A-673) (Fig. 2d). Further, genome-wide gene set enrichment analysis including every EwS cell line revealed again no significantly enriched gene ontology (GO) gene sets, canonical pathways, and protein complexes ($P < 0.05$; FDR < 0.25). Collectively, these results suggested that, while there may be subtle transcriptional changes in particular genes over time in EwS cell lines, the differences across their entire genome do not predominantly affect specific pathways or gene sets. In line with this idea, evaluation of DEGs consistency across the different EwS cell lines revealed an overlap of a single gene, mitochondrially encoded tRNA-valine (*MT-TV*) (Supplementary Fig. 2b and Supplementary Data 4), which is not an EWSR1::FLI1-signature gene[8].

We next complemented these results by exposing each newly purchased EwS cell line (m0) and their 12-month derivate (m12) to an extended drug library that contained 10 additional compounds (extended library, $n_{total} = 20$, Supplementary Data 1) to include drugs that had been recently described in EwS preclinical or clinical studies, such elesclomol, olaparib, and gemcitabine[9–11]. Here, we again

observed inter-cell line variability in collective drug response over time that ranged from the least stable A-673 to the remarkably stable TC-71 EwS cell line (Fig. 2e).

In synopsis, ranking plots for each different data layer comparing 12 months of continuous culture of each EwS cell line clearly suggest a range of stability that may inform decision-making on which cell line models to preferentially employ in this COTF-driven pediatric cancer (Fig. 2f).

Collectively, our results highlight that the findings previously described in Liu et al. and Ben-David et al. regarding the genetic and phenotypic stability of two carcinoma cell lines may not be translatable to other cancer cell lines, especially to those with a stable genetic background and a defined driver mutation, such as the COTF found in EwS (Fig. 2g). Our findings indicate that research with COTF-driven cell line models such as EwS should be in principle reproducible, even after genetic modifications, and extensive periods of continuous culture. Also, our results demonstrate that individual cell lines from the same cancer entity may display a variable degree of evolution, suggesting that the reproducibility of results strongly depends on the given cancer cell line, which is particularly relevant in the context of large-scale cell line screening efforts including Genomics of Drug Sensitivity in Cancer[12] and The Cancer Dependency Map Project[13].

## Methods

### Provenience of cell lines and cell culture conditions

For long-term culture assays the following early passage (< 5 passages) human cancer cell lines were acquired: the cervix carcinoma cell line HeLa, the human breast carcinoma cell line MCF-7. The EwS MHH-ES-1, SK-ES-1, SK-N-MC, and TC-71 cell lines were purchased from the German Collection of Microorganism and Cell Cultures (DSMZ). The A-673 EwS cell line was purchased from the American Type Culture Collection (ATCC). A-673, HeLa, and MCF-7 wild-type strains with an undefined number of passages were kindly provided by E. de Álava, U. Dirksen, K. Scotlandi, H. Kovar, I. Oehme, T. Grünewald, O. Delattre, and D. Surdez. Single-cell clones derived from A-673 cell lines with either a neutral manipulation (A-673/shcontrol) or an inducible shRNA construct against its EWSR1::FLI1 translocation (A-673/TR/shEF1) were previously described by our laboratory[8]. All cell lines were routinely tested for mycoplasma contamination by nested PCR, and cell line purity and authenticity were confirmed by STR profiling. All cell lines were cultured at 37 °C, 5% $CO_2$ in RPMI 1640 (Biochrom, Germany) supplemented with 10% fetal bovine serum (Sigma-Aldrich, Germany) and 1% penicillin-streptomycin (Merck, Germany). Each cell culture flask was monitored daily, and cells were passaged twice per week using Trypsin-EDTA (0.25%) (Life Technologies) when they reached approximately 70% confluency.

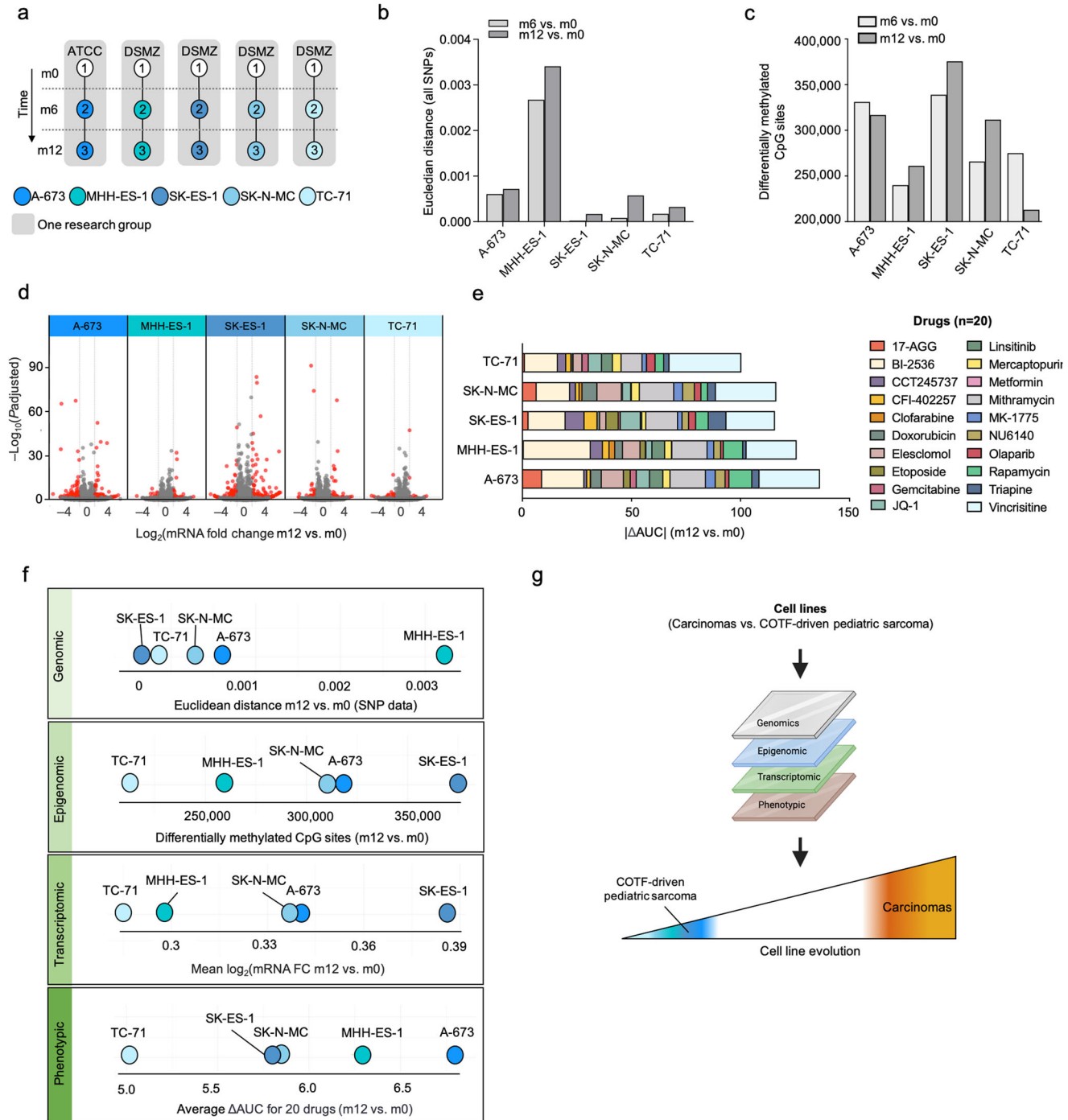

**Fig. 2 | In-depth analysis of stability on individual EwS cell lines. a** Newly purchased A-673, MHH-ES-1, SK-ES-1, SK-N-MC, and TC-71 EwS wild type cell lines (_1) were kept in culture for six months (m6; _2), and 12 months (m12; _3). ATCC, American Type Culture Collection, DSMZ (German Collection of Microorganism and Cell Cultures). **b** Bar plot shows the Euclidean distance of all SNPs after 6 (m6 vs. m0) and 12 months (m12 vs. m0) of continuous culture. **c** Number of differentially methylated CpG sites (including differentially hypo- and hyper-methylated) for A-673, MHH-ES-1, SK-ES-1, SK-N-MC, and TC-71 after six (m6) and 12 months (m12) of continuous culture, using the respective initial time point (m0) values as reference. **d** Volcano plot of DEG comparing each EwS cell line with their m12 derivate. Red dots denote significant DEG (BH adjusted $P < 0.01$; $|FC| > 1$). **e** Relative variation in cell viability for each EwS cell line measured as the mean area under the curve (AUC) m12 vs. m0 for the extended drug library (20 compounds). **f** Ranking plots depicting the linear distribution of five EwS cell lines based on their evolution across different datasets after continuous culturing for 12 months. **g** Schematic illustration summarizing the findings of this study. Created in BioRender. Aranaz, F. (2023) BioRender.com/k87h648.

## DNA extraction, methylation, and global screening arrays

When flasks reached approximately 70% confluency, samples were lysed, and total DNA was extracted with the NucleoSpin Tissue kit (Macherey Nagel) following the manufacturer's protocol. For each sample, 900 ng of DNA in one (genotyping) or two (methylation) technical replicates were used as input material and were profiled on Illumina Infinium Global Screening array and MethylationEPIC array, respectively, at the Molecular Epidemiology Unit of the German Research Center for Environmental Health (Helmholtz Center, Munich, Germany).

## Whole genome sequencing (WGS)

High-quality genomic DNA from A-673, HeLa, and MCF-7 wild-type cells at time points m0 and m6 was sequenced using the Illumina PCR-Free Tagmentation Kit (Illumina, CA, USA). A standard input of 300 ng genomic DNA was used for most samples. Sequencing was performed on the NovaSeq 6000 S4 platform using 150 bp paired-end reads. Libraries were loaded at a concentration of 200 pM with 1% PhiX control spike-in by the NGS Core Facility of the German Cancer Research Center (DKFZ, Heidelberg, Germany). WGS of A-673 wild-type cell DNA was performed as previously described (BioProject PRJNA610192)[8].

## WGS data alignment and copy number (CN) estimation

All WGS data was aligned to the hg19 reference genome using the PanCancer alignment workflow for the whole genome from the Roddy Alignment Algorithms. The aligned WGS data was used to estimate CNs with Allele-specific copy number estimation with whole genome sequencing (ACEseq) -algorithm as previously described[14]. All samples were referenced against a standardized normal control genome, which was employed because no germline tissue from the subjects was available. This normal control is derived from a pool of DNA samples from healthy individuals and serves as a reference to distinguish between somatic alterations and inherited variants. Alignment and CN estimation were done by the Omics IT and Data Management Core Facility of the DKFZ and its One Touch Pipeline[15].

## Single nucleotide variant (SNV) calling and filtering

WGS data was aligned to the hg19 reference genome using the PanCancer alignment workflow for the whole genome from the Roddy Alignment Algorithms. All samples were referenced against a standardized normal control genome derived from a pool of DNA samples from healthy individuals and serve as a reference to distinguish between somatic alterations and inherited variants. Single nucleotide variants (SNVs) were called using SNVCalling workflow from the pancancer analysis of whole genomes (PCAWG)[16]. Only high-quality (QUAL > 10) SNVs located within exonic regions of cancer-related genes were analyzed. Cancer-related genes were defined as being present in at least three of the following cancer-related gene databases: OncoKB, MSK-IMPACT, MSK-Heme, Vogelstein Cancer Genes, COSMIC CGC (v99), FoundationOne, and FoundationOne Heme[17]. Alignment and SNV calling were done by the Omics IT and Data Management Core Facility of the DKFZ and its One Touch Pipeline[15].

## WGS data alignment, copy number (CN) estimation and analysis

Aligned WGS data were used to estimate CNs with allele-specific copy number estimation with whole genome sequencing (ACEseq)-algorithm as previously described[14]. All samples were referenced against a standardized normal control genome (as described before), which was employed because no germline tissue from the subjects was available. This normal control is derived from a pool of DNA samples from healthy individuals and serves as a reference to distinguish between somatic alterations and inherited variants. CN estimation was performed by the Omics IT and Data Management Core Facility of the DKFZ and its One Touch Pipeline. WGS CN data was corrected using the batch correction algorithm from ComBat function from the sva R package version 3.50.0 (ref. 18). WGS data was then segmented into regions of estimated equal CN using the circular binary segmentation algorithm from DNAcopy R package version 1.76.0. Segmented data was used to calculate the Genomic Index (GI) as the square of the number of CN-altered DNA segments divided by the number of CN-altered chromosomes as previously described[19]. Preprocessed single nucleotide polymorphism array (Affymetrix SNP 6.0) derived CN analysis data from the Cancer Cell Line Encyclopedia (CCLE)[2] for A-673, and MCF-7 wild-type cell lines were retrieved from DepMap portal[13].

WGS of A-673 wild type derived from BioProject PRJNA610192 (ref. 8). Comparative analysis of genomic intervals between CCLE CNV data and WGS data was performed. Overlapping genomic regions were identified using the findOverlaps function from the IRanges R package version 2.36.0 (ref. 20). The filtering criteria included the following conditions: the start position of the CCLE genomic interval must be less than or equal to the end position of the WGS genomic interval, and the end position of the CCLE genomic interval must be greater than or equal to the start position of the WGS genomic interval. In addition, the matching interval of the WGS data had to be 80%–120% of the CCLE interval's size. Subsequently, the values of overlapping WGS intervals within each CCLE interval were aggregated by calculating the mean CN for all overlapping WGS intervals. The area under the curve of CN ratios was calculated using Graphpad PRISM 9, v9.4.1.

## DNA methylation data analysis

The initial pre-processing of the raw methylation was performed in R version 3.3.1. Raw signal intensities were obtained from IDAT files using the minfi Bioconductor package version 1.21.4[21] in R version 3.3.1. Each sample was individually normalized by performing a background correction (shifting of the 5% percentile of negative control probe intensities to 0) and a dye-bias correction (scaling the mean of normalization control probe intensities to 10,000) for both color channels. The methylated and unmethylated signals were corrected individually. Subsequently, beta values were calculated from the retransformed intensities using an offset of 100 (as recommended by Illumina). Out of 865,859 probes on the EPIC array, 105,454 probes were masked, according to Zhou et al. [22] as well as 16,944 probes on the X and Y chromosomes. In total, 743,461 probes were kept for downstream analysis. The beta values were transformed to M-values with the logit2 function of the *minfi* package version 1.42.0, R version 4.2.0. A probe-wise differential methylation analysis[23] was performed using the *limma* package[24] version 3.52.4 in R version 4.2.0 by comparing six and twelve months of culturing with the initial time point (m0) as reference. Significant differentially methylated CpG probes were extracted with the *decideTests* function of the *limma* package with an FDR < 0.05 (Benjamini-Hochberg). All significantly differentially methylated (total hypo- and hyper-methylated) CpG sites were visualized using PRISM 9 (GraphPad Software Inc. CA, USA). Differentially methylated promoter regions (DMPRs) were identified by encompassing CpG sites within promoter regions defined using the mCSEA package, version 1.16.0 in R version 4.2.0. Differential methylation analysis of promoter regions was performed by aggregating CpG sites into promoter regions and calculating average methylation levels. A region-wise differential methylation analysis was conducted using the minfi package to identify regions with significant differential methylation. Statistical significance for promoter regions was determined with an FDR < 0.05 (Benjamini-Hochberg correction). Only promoter regions containing at least five CpG sites were considered for this analysis (default setting of the mCSEATest function).

## Global screening array (GSA) data analysis

The initial processing and quality control (QC) of the raw genotyping data was performed using PLINK version 1.9 (SNP call rate > 95%, Hardy-Weinberg exact test < 1e-6, and variants on the Y chromosome were excluded). In total 526,610 variants out of 696,726 passed the QC filters. Infinium GSA v3.0 annotation file was used to filter for in-exon or non-synonymous variants. To determine single nucleotide alterations (SNA), A-673 strains were compared to its m0 version (number of consistent alleles and changes from homozygous to heterozygous) using the Variant Call Format (VCF) file generated by PLINK 1.9. Further data analysis was performed in R version 4.2.1, using the vcfR package version 1.14.0, among other data processing packages described below. The distance between two-time points for each cell line was computed in R version 4.2.1 using the proxy package version 0.4-27.

The eigenvectors generated for dimension reduction in PLINK version 1.9, were used as input. The heatmap was generated in R version 4.2.1 using the pheatmap package version 1.0.12.

### RNA extraction, library preparation, RNA sequencing and analysis

When flasks reached ~70% confluency, total RNA was isolated using the NucleoSpin RNA kit (Macherey-Nagel, Germany) according to the manufacturer's protocol. RNA quality was verified on a Nanodrop Spectrophotometer ND-1000 (Thermo Fischer), and quantity was measured on a Qubit instrument (Life Technologies). For each sample, 50–100 ng of RNA in three biological and two technical replicates were used as input material and were profiled on an Illumina NextSeq 500 system at the Institute of Molecular Oncology and Functional Genomics in Rechts der Isar University Hospital (TranslaTUM Cancer Center, Munich, Germany). Library preparation for bulk 3′-sequencing of poly(A)-RNA was performed as previously described[25]. Briefly, the barcoded cDNA of each sample was generated with a Maxima RT polymerase (Thermo Fisher) using oligo-dT primer containing barcodes, unique molecular identifiers (UMIs), and an adapter. 5′ ends of the cDNAs were extended by a template switch oligo (TSO), and after pooling of all samples full-length cDNA was amplified with primers binding to the TSO-site and the adapter. cDNA was fragmented, and TruSeq-Adapters ligated with the NEBNext® Ultra™ II FS DNA Library Prep Kit for Illumina® (NEB), and 3′-end-fragments were finally amplified using primers with Illumina P5 and P7 overhangs. P5 and P7 sites were exchanged to allow sequencing of the cDNA in read1 and barcodes and UMIs in read2 to achieve better cluster recognition. The library was sequenced with 75 cycles for the cDNA in read1 and 16 cycles for the barcodes and UMIs in read2. Data was processed using the published Drop-seq pipeline (v1.0) to generate sample- and gene-wise UMI tables[26]. After the elimination of transcripts with very low counts (sums of all samples < 10), RNASeq data in count matrix format was batch corrected using the ComBat-Seq function of R package sva version 3.44.0 (ref. 18), and differential gene expression analysis (DGEA) was performed using DESeq2 version 1.36.0 (ref. 27) on R version 4.2.1. Combat-Seq adjusted data was used as count input for DESeqDataSet. To avoid false discovery artifacts due to the detection of minimally expressed genes, we excluded the 40% lowest expressed genes across samples (remaining expressed genes $N = 10{,}257$). For the analysis of long-term cultured EwS cell lines we performed DGEA on the top 60% expressed genes included in the raw count matrix ($N = 27{,}143$, all EwS cell line samples were analyzed in one batch). For DGEA between two samples, genes with $P_{adj} \leq 0.01$, $|log2(FC)| > 1$ were considered as DEG. Principal component analysis was used to preserve the global properties of the data using the *plotPCA* function. To comprehensively display the degree of variability between strains in each tumor type, the gene-specific CV of the transcriptomic data was calculated. In the long-term culture assays, $log_2FC$ of gene expression of each cell line for 6 and 12 months (m6 and m12) were analyzed using the initial time point (m0) values as reference.

### Drug screening

All A-673, HeLa, and MCF-7 strains, as well as MHH-ES-1, SK-ES-1, SK-N-MC, and TC-71 EwS cell lines, were tested against a core drug library consisting of 10 cytotoxic or cytostatic agents, or an extended drug library consisting of 20 agents (Supplementary Data 1). For this, cells were seeded into 96-well plates at a density of $5 \times 10^3$ cells per well in 90 µl of medium in triplicates. Once cells were attached, ~4 h after seeding, 10 µl of each compound was added in serially diluted concentrations ranging from $1 \times 10^{-5}$ µM to 10 µM. DMSO was used as vehicle control. Plates were incubated for 72 h at 37 °C, with 5% $CO_2$ in a humidified atmosphere. At the experimental endpoint, a solution of 25 µg/ml of resazurin salt (Sigma-Aldrich) was added to the medium,

and cell viability was determined as previously described[28]. Each compound and cell line were assayed in four biological replicates.

### Drug screening data analysis

Cell viability data was first normalized using the measured raw viability of each control (DMSO vehicle), and the area under the curve (AUC) was computed for each cell line using the PharmacoGx package version 3.0.2 (P Smirnov, 2016) in R version 4.2.1. Euclidean distances (ED) between drug sensitivity profiles of each strain were calculated using the following formula:

$function(x1, x2) \, sqrt(sum((x1 - x2)^2)) = ED$, where x1 is the mean value of AUC of all strains and x2 the AUC of individual cell lines. The variability in drug response across different cancer entities was visualized using the standard error of ED values, accounting for differences in sample size. Changes in drug sensitivity during the long-term culture of each cell line for six and 12 months (m6 and m12) were analyzed using the initial time point (m0) values as reference.

### Other bioinformatic and statistical analyses

If not otherwise specified, genomic, methylation, transcriptomic, and drug sensitivity data analyses were performed in R version 4.2.1. The following R packages were used: for data processing, readxl package version 1.4.3, tidyverse package version 2.0. (ref. 29), reshape2 package version 1.4.4 (ref. 30), cowplot package version 1.1.1, Rfast package version 2.0.8, and data.table package version 1.14.8 (ref. 31); for data visualization, ggplot2 package version 3.4.1 (ref. 32), gghalves package version 0.1.4, ggdist package version 3.2.1 and PupillometryR package version 0.0.4; for circle plots, circlize package version 0.4.15; and for PCA and volcano plots, ggplot2 package version 3.4.1 (ref. 32). Spearman's correlation analyses of quantitative data of both mRNA and drug response were performed using Hmisc package version 4.7-2 (ref. 33). Figures 1b, 1i, 1j, 2b, 2c, 2e, and Supplementary Fig. 1b were generated using PRISM 9 (GraphPad Software Inc., Ca, USA). Transcriptomic datasets from this study and Liu et al.[5] were combined and batch-corrected using the ComBat-Seq function of package sva version 3.44.0 (ref. 34). Venn diagrams were plotted using Affinity Designer 2, version 2.4.2.

### Reporting summary

Further information on research design is available in the Nature Portfolio Reporting Summary linked to this article.

## Data availability

Original data that support the findings of this study was deposited at the National Center for Biotechnology Information (NCBI) GEO under accession numbers GSE270195, GSE268437, GSE264509, and under BioProject PRJNA1160032. Source data are provided in this paper.

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

## Acknowledgements

M.K. received scholarships from the German Cancer Aid ('Mildred-Scheel-Doctoral Program') and the Rudolf und Brigitte Zenner Stiftung. F.H.G. was supported by the German Academic Scholarship Foundation and the German Cancer Aid through the 'Mildred-Scheel-Doctoral Program'. The research team of F.C.A. was supported by the German Cancer Aid (DHK-70114111), the Dr. Rolf M. Schwiete Stiftung (2020-028 and 2022-31) and Cancer Grand Challenge, Cancer Research UK (PROTECT). The laboratory of T.G.P.G. was supported by the Matthias-Lackas Foundation, Dr. Leopold and Carmen Ellinger Foundation, the German Cancer Aid (DKH-70112257, DKH-70114278, DKH-70115315), the SMARCB1 association, the Federal Ministry of Education and Research (BMBF-projects SMART-CARE and HEROES-AYA), the Deutsche Forschungsgemeinschaft (DFG-458891500), and the Barbara and Wilfried Mohr Foundation. This project is co-funded by the European Union (ERC, CANCER-HARAKIRI, 101122595). Views and opinions expressed are however those of the authors only and do not necessarily reflect those of the European Union or the European Research Council. Neither the European Union nor the granting authority can be held responsible for them. We thank Prof. Heinrich Kovar for kindly sharing materials, and Dr. Soledad Gómez-Gonzalez for critical discussion of this manuscript. We thank the NGS Core Facility, and the Omics IT and Data Management Core Facility (ODCF) of the German Cancer Research Center (DKFZ) for providing excellent WGS and data management services.

## Author contributions

F.C.A. and T.G.P.G. conceived the study. M.K. performed all experiments and bioinformatic and statistical analyses. F.C.A. contributed to drug screening experiments. F.H.G., J.S., and T.F. contributed to bioinformatic analyses. M.S., R.Ö., R.R., M.M-N., and K.St. contributed to sample analysis and/or provided laboratory infrastructure. E.deÁ., D.S., O.D., U.D., I.Ö., and K.Sc. provided cell line models. M.K., F.C.A., and T.G.P.G. wrote the paper and drafted the figures and tables. F.C.A and T.G.P.G. supervised the study and data analysis. All authors read and approved the final manuscript.

## Funding

## Competing interests

The authors declare no competing interests.

## Additional information

[1]Hopp Children's Cancer Center (KiTZ), Heidelberg, Germany. [2]Division of Translational Pediatric Sarcoma Research (B410), German Cancer Research Center (DKFZ), German Cancer Consortium (DKTK), Heidelberg, Germany. [3]National Center for Tumor Diseases (NCT), NCT Heidelberg, a partnership between DKFZ and Heidelberg University Hospital, Heidelberg, Germany. [4]Max-Eder Research Group for Pediatric Sarcoma Biology, Institute of Pathology, Faculty of Medicine, LMU Munich, Munich, Germany. [5]Division of Pediatric Neurooncology, German Cancer Research Center (DKFZ), Heidelberg, Germany. [6]Transla-TUM, Center for Translational Cancer Research, Technical University of Munich, Munich, Germany. [7]Institute of Biomedicine of Sevilla (IBiS), Virgen del Rocio University Hospital/CSIC/University of Sevilla/CIBERONC, Seville, Spain. [8]Department of Normal and Pathological Cytology and Histology, School of Medicine, University of Seville, Seville, Spain. [9]INSERM U830, Diversity and Plasticity of Childhood Tumors Lab, PSL Research University, SIREDO Oncology Center, Institut Curie Research Center, Paris, France. [10]Department of Pediatrics, University Hospital Essen, Essen, Germany. [11]Clinical Cooperation Unit Pediatric Oncology, German Cancer Research Center (DKFZ) and German Cancer Consortium (DKTK), Heidelberg, Germany. [12]Experimental Oncology Laboratory, IRCCS Istituto Ortopedico Rizzoli, Bologna, Italy. [13]Institute of Medical Biostatistics, Epidemiology and Informatics (IMBEI), University Medical Center, Johannes Gutenberg University, Mainz, Germany. [14]IBE, Faculty of Medicine, LMU Munich, Munich, Germany. [15]Institute of Genetic Epidemiology, Helmholtz Zentrum München - German Research Center for Environmental Health, Neuherberg, Germany. [16]Department of Medicine II, Klinikum Rechts der Isar, Technical University Munich, Munich, Germany. [17]German Cancer Consortium (DKTK), German Cancer Research Center (DKFZ), Munich, Germany. [18]Institute of Pathology, Heidelberg University Hospital, Heidelberg, Germany. [19]Present address: Balgrist University Hospital, Faculty of Medicine, University of Zurich (UZH), Zurich, Switzerland. ✉e-mail: florencia.cidrearanaz@dkfz.de

