## [Transparent Peer Review file · Nature Communications]

Genomic and phenotypic stability of fusion-driven pediatric sarcoma cell lines

Corresponding Author: Dr Florencia Cidre-Aranaz

Version 0:

Reviewer comments:

Reviewer #1

(Remarks to the Author)

Kasan et al report an interesting study which contributes to our understanding of what happens to cell lines in culture. I don't have major issues with the paper technically apart from their approach to genotyping. Really this should have been whole genome sequencing with time point 0 being used as the reference to call variants against. Another interesting assay would have been single cell mRNA sequencing which I suspect would not have changed with time. Bulk mRNAseq is a little outdated these days. Overall this is clearly publishable work. Whether or not Nat Comms is the right journal is not my call.

Reviewer #2

(Remarks to the Author)

Prior studies from Ben-David et al. and Liu et al., suggested that adult carcinoma lines demonstrate striking diversity across laboratories and with long-term continuous culturing leading to changes in their genomics/transcriptomics and response to perturbations, including drug response. Such findings have implications for the interpretation of data generated with cancer cell lines. Whether this is the case, however, for cancer cell lines driven by a key fusion oncoprotein with few recurrent point mutations is not known. This study by Kasan et al. demonstrates that cell lines derived from Ewing sarcoma, a pediatric cancer driven by the EWS-FLI1 fusion oncoprotein, are much more genomically and transcriptionally stable than adult carcinoma strains and exhibit a more homogeneous response to drug perturbations across strains or continuous culture. This is a well written and an important study to publish as it has highly relevant findings to the field of cancer research, including large scale projects such as Sanger Cell Line Drug Response, Dependency Map, etc. There are a few points that should be addressed before publication of this manuscript.

1) The authors use Global Screening Arrays and conduct SNP analysis. The prior studies by Ben-David/Liu also used copy number analysis. It would be important to also evaluate copy number, which can be done with the GSA data.

2) What are the genes harboring the variable ns-SNPs across the strains, especially in MHHES1 and A673 cells? Are they enriched for oncogenes/tumor suppressors, EWS-FLI1 target genes, or other expressed genes?

3) The PCA plot in Fig1d captures the high variability among the cell line lineages (EwS/breast cancer/cervical cancer) rather than the variability within each lineage. The authors might analyze the A673 stability based on a PCA plot or a genome-wide dendrogram restricted to only the A673 cell line strains. What are the leading-edge genes in the principal component loadings for the EwS strains?

4) What are the top differentially expressed genes for the EwS cell lines at m0 versus m12? Are they consistent across the EwS cell lines? Are they enriched for any pathways or features (EWS-FLI1 target genes, etc).

5) A genome-wide gene set enrichment analysis comparing m12 to m0 (applied to all "expressed" genes in the genome, not only to the top 60% genes with higher expression) may reveal functional changes that are induced by subtle but coherent transcriptional differences. What are the differentially enriched gene ontology/canonical pathways/protein complexes gene sets in the EwS strains at m12 vs m0?

6) Please provide more details regarding the consistency of the differentially methylated CpG sites across the EwS cell lines

depicted in Fig 2c?

Version 1:

Reviewer comments:

Reviewer #1

(Remarks to the Author)

The authors have addressed my concerns adequately.

Reviewer #2

(Remarks to the Author)

The authors have successfully addressed my questions. I believe that this manuscript is of high quality and impact and should be published in Nature Communications.

Response to the Reviewers:

We would like to thank the Reviewers for their time and dedication to review our manuscript and for the constructive feedback and questions provided. We have addressed all points raised in full, and we appreciate that these revisions significantly improved the manuscript. Relevant changes to the manuscript text have been highlighted in blue font.

Reviewer #1 (Remarks to the Author):

Kasan et al report an interesting study which contributes to our understanding of what happens to cell lines in culture. I don't have major issues with the paper technically apart from their approach to genotyping. Really this should have been whole genome sequencing with time point 0 being used as the reference to call variants against. Another interesting assay would have been single cell mRNA sequencing which I suspect would not have changed with time. Bulk mRNAseq is a little outdated these days. Overall this is clearly publishable work. Whether or not Nat Comms is the right journal is not my call.

Authors: We thank this Reviewer for the positive evaluation of our paper from a technical perspective and for pointing out that this work is highly relevant for publication.

We agree with the notion that WGS would add a more comprehensive view to the genotyping analyses and variant calling. To accommodate this suggestion, we performed WGS on A-673, MCF-7 and HeLa cell lines corresponding to month 0 and 6, and used in each case the corresponding m0 strain as the reference to call variants against as suggested. The new results of these analyses are depicted in the new **Fig. 1b**, and are now described in the manuscript in lines 349-354 as follows:

“In a first step, we performed a cross-strain analysis of A-673, MCF-7, and HeLa cell lines and subjected each newly purchased cell line (m0) and its respective m6-cultured version to whole genome sequencing (WGS), which enabled us to monitor genetic evolution over time. Analysis of relative in-exon SNVs counts in cancer genes for A-673, HeLa, and MCF-7 after six months of continuous culture revealed a general stability of A-673 strains as compared to HeLa and MCF-7 cells (**Fig. 1b**).”

In addition to these analyses, and to better understand the potential genomic variabilities resulting from copy number alteration in strains grown in different laboratories (and with potentially different number of passages/culture conditions), we took advantage of already available WGS from A-673 included our Ewing Sarcoma Cell Line Atlas (ESCLA) project that was published earlier (Orth et al, 2022), and datasets from the Cell Line Encyclopedia (CCLE) for A-673 and MCF-7, and compared them to our new WGS results for the A-673_m0 and MCF-7_m0 strains. As shown in the new Supp. Figs. 1a,b, differential copy number analyses revealed that – as expected – MCF-7 strains display a strong variability in copy numbers. However, the three A-673 strains exhibited remarkably low variability in CN as analyzed by CNV ratio between them, and area under the curve of each CNV ratio.

These exciting results are now displayed in the new **Supp. Fig. 1a,b**, and in the manuscript in lines 354-362 as follows:

“To explore the differences in genome stability comparing COTF-driven strains to carcinoma strains in more detail, we employed WGS data and compared copy number alternations of A-673 and MCF-7 strains from different laboratories including our A-673_m0 and MCF-7_m0, those of the Cancer Cell Line Encyclopedia (CCLE), and an A-673 strain from the Ewing Sarcoma Cell Line Atlas (ESCLA). As observed in **Supp. Fig. 1a,b**, the A-673 strains generally presented a more stable genome compared to MCF-7 strains, as quantified by relative changes in copy numbers.”

Given the only very minimal transcriptomic changes over time of our tested EwS cell lines as detected by bulk-level RNA-seq, we fully concur with the notion of this Reviewer that single-cell RNA-seq analyses would very likely not have contributed any additional insights to this manuscript. Given that also Reviewer #2 did not make any recommendation in this regard, we did not conduct such analyses at this stage.

Reviewer #2 (Remarks to the Author):

Prior studies from Ben-David et al. and Liu et al., suggested that adult carcinoma lines demonstrate striking diversity across laboratories and with long-term continuous culturing leading to changes in their genomics/transcriptomics and response to perturbations, including drug response. Such findings have implications for the interpretation of data generated with cancer cell lines. Whether this is the case, however, for cancer cell lines driven by a key fusion oncoprotein with few recurrent point mutations is not known. This study by Kasan et al. demonstrates that cell lines derived from Ewing sarcoma, a pediatric cancer driven by the EWS-FLI1 fusion oncoprotein, are much more genomically and transcriptionally stable than adult carcinoma strains and exhibit a more homogeneous response to drug perturbations across strains or continuous culture. This is a well written and an important study to publish as it has highly relevant findings to the field of cancer research, including large scale projects such as Sanger Cell Line Drug Response, Dependency Map, etc.

Authors: We thank this Reviewer for highlighting the high relevance of our findings for the field of cancer research, the importance of its derived conclusions, as well as the clarity of our manuscript. We also thank this Reviewer for pointing out the broad relevance of our manuscript in frame of several large-scale projects including Sanger Cell Line Drug Response, Dependency Map, etc. We agree that this is an important aspect to highlight, so we have now modified the Conclusion section and added relevant references to the suggested large-scale projects as follows (lines 492-494):

“Also, our results demonstrate that individual cell lines from the same cancer entity may display a variable degree of evolution, suggesting that the reproducibility of results strongly depends on the given cancer cell line, which is particularly relevant in the context of large-scale cell line screening efforts including Genomics of Drug Sensitivity in Cancer³⁴ and The Cancer Dependency Map Project¹⁵.”

There are a few points that should be addressed before publication of this manuscript. 1) The authors use Global Screening Arrays and conduct SNP analysis. The prior studies by Ben-David/Liu also used copy number analysis. It would be important to also evaluate copy number, which can be done with the GSA data.

Authors: We thank this Reviewer for pointing out this important aspect. As this Reviewer suggests, it is in fact possible to perform this analysis using GSA-derived data that we had already generated, although this approach has some limitations. To overcome these, we have additionally performed WGS (as suggested by Reviewer #1, for a description of the experimental specifics, please see the response to Reviewer #1). As a result, we have now included the new **Supp. Figs. 1a,b** displaying that, as expected, MCF-7 strains display strong variability in CNs. However, the three A-673 strains analyzed showed remarkably low variability in CN as analyzed by CNV ratio between each pair, and area under the curve of each CNV ratio. The corresponding text has been adapted as follows (lines 354-362):

“To explore the differences in genome stability comparing COTF-driven strains to carcinoma strains in more detail, we employed WGS data and compared copy number alternations of A-673 and MCF-7 strains from different laboratories including our A-673_m0 and MCF-7_m0, those of the Cancer Cell Line Encyclopedia (CCLE), and an A-673 strain from the Ewing Sarcoma Cell Line Atlas (ESCLA). As observed in **Supp. Fig. 1a,b**, the A-673 strains generally presented a more stable genome compared to MCF-7 strains, as quantified by relative changes in copy numbers.”

2) What are the genes harboring the variable ns-SNPs across the strains, especially in MHHES1 and A673 cells? Are they enriched for oncogenes/tumor suppressors, EWS-FLI1 target genes, or other expressed genes?

Authors: We thank this Reviewer for raising these important questions. To address them, we examined ns-SNPs in our 11 A-673 and three MHH-ES1 samples (strains and/or long-term cultured cell lines). We then identified those ns-SNPs with a varying SNP status (homozygous for the reference or alternate allele, or heterozygous) in at least one strain for each cell line. Next, we annotated those SNPs IDs to their corresponding genes as detailed in new **Suppl. Table 2** of our revised manuscript. Each gene list was further employed to explore their potential enrichment in oncogenes/tumor suppressors, EWS-FLI1 target genes, or other expressed genes. As a first approach, when each gene list was analyzed for their significant enrichment in Gene Ontology biological processes (Fisher's exact test, correction by FDR), no specific process was identified. Similarly, no significantly enriched cellular component, molecular function or pathway was identified (Fisher's exact test, correction by FDR).

To explore the presence of EWS-FLI1 targets on the variable ns-SNPs-containing genes for A-673 and MHH-ES1, we analyzed our previously described EWS-FLI1-signature genes (Orth et al., 2022). For MHH-ES1, our analysis revealed no overlap. In the case of A-673, there was only one hit, corresponding to the UNC5 family of netrin receptors (*UNC5B*), a gene involved in mediating cell migration and apoptosis. It should be however noticed that this gene was not found to be differentially expressed at the transcriptomic level in A-673 ($|\log_{2}FC| > 1$, $P_{adj} < 0.05$).

In summary, these new analyses indicate that the variable ns-SNPs in EwS cell lines generally do not show a significant enrichment for oncogenes, tumor suppressors, or EWS-FLI1 target genes. The relatively low number of variable ns-SNPs found in EwS cell lines affect genes that appear to be randomly distributed and not centered around specific pathways, biological

processes or essential molecular functions of EwS cells, including those genes regulated by the EWS-FLI1 fusion.

To acknowledge these important observations in the manuscript, we have now added a new **Suppl. Table 2**, and the following text in lines 448-451:

“Of note, the relatively low number of variable ns-SPNs found in A-673 strains appeared affect random genes, and to be not enriched in specific pathways or biological processes (Supp. Table 2). Only one known EWSR1::FLI1-signature gene (UNC5 family of netrin receptors, UNC5B)⁸ was affected.”

3) The PCA plot in Fig1d captures the high variability among the cell line lineages (EwS/breast cancer/cervical cancer) rather than the variability within each lineage. The authors might analyze the A673 stability based on a PCA plot or a genome-wide dendrogram restricted to only the A673 cell line strains. What are the leading-edge genes in the principal component loadings for the EwS strains?

Authors: We thank this Reviewer for bringing this important aspect to our notice, which would indeed help the reader better understand the precise variability within each lineage. To address this in more detail, we conducted independent differential gene expression analyses (DGEA) and principal component analysis (PCA) on the 11 A-673, five HeLa, and five MCF-7 strains and computed the variance percentages across each cell line’s strains. As depicted in the new **Suppl. Fig. 1c**, the A-673 strains demonstrated a 2- to 3-fold smaller variance compared to the carcinoma cell lines, again highlighting the higher stability of the A673 strains, even considering the larger sample size in the A-673 collection. These new aspects are now described in the manuscript in lines 378-383 and new **Supp. Fig. 1c**:

“To analyze this variability specifically within each lineage, we conducted independent DGEA and PCA on the 11 A-673, five HeLa, and five MCF-7 strains and computed the variance percentages across each cell line’s strains. We thus observed that the A-673 strains demonstrated a 2- to 3-fold smaller variance compared to the carcinoma cell lines, again highlighting the higher stability of the A673 strains, even considering the larger sample size in the A-673 collection (**Suppl. Fig. 1c**).”

In addition, to explore the specific genes driving the relatively small variability in A-673 strains, we performed DEG analysis on the 11 strains using DESeq2 and identified 100 DEGs (as defined by $|\log_{2}FC| > 1$, Benjamini-Hochberg (BH)-adjusted $P < 0.01$). As suggested by this Reviewer, to better understand if the DEGs driving the PCA loadings were enriched in specific biological processes that could give first clues on preferential cell programs involved in the evolution of EwS cell lines, we performed FDR-corrected GO analysis. This revealed significant enrichment in exclusively one biological process: ‘cholesterol biosynthetic process’. Cholesterol biosynthesis in cancer is typically associated with higher proliferation rates since cholesterol is essential for plasma membrane generation. However, we did not detect any enrichment in proliferation associated signatures. Thus, the biological meaning of this finding remains elusive at this stage, which is why we would prefer to not speculate about this aspect and thus prefer to not address it in the manuscript at the present time.

4) What are the top differentially expressed genes for the EwS cell lines at m0 versus m12? Are they consistent across the EwS cell lines? Are they enriched for any pathways or features (EWS-FLI1 target genes, etc).

Authors: We thank this Reviewer for requesting more information on this intriguing topic. To address this, we re-examined our resulting DEGs for each EwS cell line at m0 versus m12 present in Figure 2d and explored the intersection among them in all EwS cell lines. As it can now be observed in the new **Suppl. Fig. 2b** and new **Suppl. Table 4**, the vast majority of DEGs are not shared among EwS cell lines. In fact, these analyses revealed only one gene, mitochondrially encoded tRNA-Valine (*MT-TV*), as consistently differentially expressed across all EwS cell lines. It should be noted that *MT-TV* is not included in our previously described EWS-FLI1-signature (Orth et al., 2022). Although *MT-TV* has been linked to mitochondrial dysfunction (Arredondo *et al.*, 2012), its precise function in EwS is unknown. However, its consistent upregulation upon continuous culture of all EwS cell lines studied here may suggest a common response to mitochondrial stress or an adaptation response to impaired mitochondrial function.

To address these aspects in the manuscript we have now included a new **Suppl. Fig. 2b** and new **Suppl. Table 4**, and the following text in lines 466-469:

“In line with this idea, evaluation of DEGs consistency across the different EwS cell lines revealed an overlap of a single gene, mitochondrially encoded tRNA-Valine (*MT-TV*) (**Suppl. Fig. 2b, Suppl. Table 4**), which is not an EWSR1::FLI1-signature gene⁸ and whose precise function on EwS is unknown..”

5) A genome-wide gene set enrichment analysis comparing m12 to m0 (applied to all “expressed” genes in the genome, not only to the top 60% genes with higher expression) may reveal functional changes that are induced by subtle but coherent transcriptional differences. What are the differentially enriched gene ontology/canonical pathways/protein complexes gene sets in the EwS strains at m12 vs m0?

Authors: We thank this Reviewer for this excellent suggestion. To address this aspect, we performed a genome-wide analysis comparing m12 vs m0 in all EwS cell lines and including all “expressed” genes –and not just the top 60% with higher expression. Despite a thorough enrichment analysis of gene ontology (GO) terms, canonical pathways, and protein complexes, we did not identify any statistically significant enriched gene sets ($P < 0.05$; $FDR < 0.25$). This suggests that while there may be subtle transcriptional changes in particular genes over time in EwS cell lines, the differences across their entire genome do not predominantly affect specific pathways or gene sets.

We have now included this aspect in the manuscript as follows in lines 460-466:

“Further, genome-wide gene set enrichment analysis including every EwS cell line revealed again no significantly enriched gene ontology (GO) gene sets, canonical pathways, and protein complexes ($P < 0.05$; $FDR < 0.25$). Collectively, these results suggested that, while there may be subtle transcriptional changes in particular genes over time in EwS cell lines, the differences across their entire genome do not predominantly affect specific pathways or gene sets.”

6) Please provide more details regarding the consistency of the differentially methylated CpG sites across the EwS cell lines depicted in Fig 2c?

Authors: We apologize that this aspect was possibly explained with insufficient detail. To address the consistency of the differentially methylated CpG sites across the EwS cell lines focusing on the predominantly regulatory sites for CpG islands, we performed a differential methylation analysis of the promoter regions for each gene using each EwS cell line at m12 versus m0. Here we observed an overlap in exclusively 51 promoter regions (1% of all promoter regions analyzed, see new **Supp. Fig. 2a**, and new **Supp. Table 3**).

The results of this analysis are detailed in the new **Supp. Figure 2a** and **Supp. Table 3** of our revised manuscript, and in the following text in lines 451-454:

“...Additionally, when we tested the consistency of differentially methylated CpG sites across all EwS cell lines particularly located at promoter regions, we found only 1% overlap (corresponding to 51 promoter regions) (**Supp. Fig. 2a**, **Supp. Table 3**).”

REVIEWERS' COMMENTS

Reviewer #1 (Remarks to the Author):

The authors have addressed my concerns adequately.

Reviewer #2 (Remarks to the Author):

The authors have successfully addressed my questions. I believe that this manuscript is of high quality and impact and should be published in Nature Communications.

Authors: We sincerely thank both reviewers for their positive evaluation of our revised manuscript, and for their comments and questions that have undoubtedly enriched our final manuscript.